# Dynamic modeling of mortality risk factors in Ebola virus disease using logistic regression on unbalanced panel data from a randomized controlled trial in the Democratic Republic of Congo

**Leader Lawanga Ontshick**[1,2]*, **Jepsy Yango**[1*], **Ange Mubiala Yaya**[3],
**Olivier Tshiani Mbaya**[3,4], **Joule Madinga Twan**[1], **Jean-Michel Nsengi Ntamabyaliro**[5],
**Rosine Ali**[6,7], **Patrick Mutombo Lupola**[1], **Joseph-Desiré Bukweli**[2,]
**Sifa Marie-joelle Muchanga**[8,9], **Gaston Tona Lutete**[5], **Placide Mbala Kiangebeni**[1,4],
**Sabue Mulangu**[3,4,10], **Rostin Mabela Makengo Matendo**[2]

**1** Department of Epidemiology and Global Health, National Institute of Biomedical Research, Kinshasa, Democratic Republic of Congo, **2** Department of Mathematics, Statistics and Computer Science, University of Kinshasa, Kinshasa, Democratic Republic of Congo, **3** Immunology Department, National Institute of Biomedical Research, Kinshasa, Democratic Republic of Congo, **4** Department of Medical Biology, University of Kinshasa, Kinshasa, Democratic Republic of Congo, **5** Department of Pharmacology and Therapeutics, University of Kinshasa, Kinshasa, Democratic Republic of Congo, **6** Department of Parasitology, National Institute of Biomedical Research, Kinshasa, Democratic Republic of Congo, **7** Department of Biology, University of Kinshasa, Kinshasa, Democratic Republic of Congo, **8** Department of Obstetrics and Gynecology, Kinshasa, Democratic Republic of the Congo, **9** Department of International Trials, National Center for Global Health and Medicine, Tokyo, Japan, **10** Ridgeback Biotherapeutics, Miami, Florida, United States of America

☯ These authors contributed equally to this work.
* leader.ontshick@gmail.com (LLO); jepsyango@gmail.com (JY)

## Abstract

Ebola Virus Disease (EVD) remains a significant public health threat, particularly in sub-Saharan Africa. During the 10th Ebola outbreak in the Democratic Republic of Congo (DRC), the Pamoja Tulinde Maisha clinical trial (PALM-RCT) provided a unique opportunity to evaluate new therapeutic interventions. Despite these advances, limited knowledge exists regarding the dynamic evolution of mortality risk factors in EVD patients. This study aimed to model risk factors associated with mortality using logistic regression on unbalanced panel data from patients enrolled in this trial.We conducted a retrospective secondary analysis of longitudinal data from 617 EVD patients included in the PALM-RCT. Data were collected at five time points: Day0 (admission), Day7, Day14, Day21, and Day28. A binary logistic regression model was applied at each time point to identify significant predictors of mortality. The Hosmer-Lemeshow test was used to assess model calibration and internal validation. At Day0 (admission), six significant predictors of mortality were identified: viral load (RT-PCR cycle threshold value), creatinine, alanine aminotransferase (ALAT), aspartate aminotransferase (ASAT), haemorrhage, shortness of breath, and conjunctivitis.

**Data availability statement:** All relevant data supporting the findings of this study are included in the manuscript and its Supporting Information files. These materials are available for download and review.

**Funding:** The author(s) received no specific funding for this work.

**Competing interests:** The authors have declared that no competing interests exist.

By Day7, five predictors emerged: sodium, ASAT, coma, abdominal pain, and shortness of breath. At Day14, two predictors remained significant: ASAT and mental state changes. No significant predictors were identified at Day21 and Day28. The dynamic nature of these risk factors highlights the importance of continuous monitoring throughout the clinical course of EVD.Our study demonstrates that mortality risk factors in EVD patients evolve over time, suggesting that a dynamic approach to patient monitoring is critical. Early risk factors such as viral load and renal function should guide initial interventions, while neurological symptoms and electrolyte imbalances require attention in later stages. These findings support a personalized approach to EVD management, where clinical care is adjusted based on real-time clinical data to improve patient outcomes.

## Introduction

The Ebola virus epidemic (EVD) remains a major threat to global public health, particularly in sub-Saharan Africa. The 10th Ebola epidemic in the Democratic Republic of Congo (DRC), which began in 2018, was one of the most serious, both in terms of the number of cases and the complexity of the response to the crisis [1]. Clinical trials conducted during this period have assessed the efficacy of new treatments and vaccines, offering promising prospects for controlling this highly lethal disease [2].

While several studies have explored risk factors associated with mortality in EVD outbreaks, most of these analyses rely on data collected at a single time point, typically at patient admission. This static approach limits the understanding of how risk evolves throughout the course of illness. There remains a critical need for analytical methods that can capture the dynamic progression of disease and changes in risk factors over time especially in clinical trial settings where longitudinal data are available but often underutilized [2,3].

Analysis of risk factors for Ebola virus-related mortality in clinical trials is essential to optimize clinical care including therapeutic interventions and improve survival rates [4]. However, most studies focus only on data from patient inclusion ($Day_0$), which limits the ability to understand the impact of changes in patients' clinical status over time [2,5–8]. Given the longitudinal data collected during phase II clinical trials, it is crucial to develop analytical methods capable of considering the temporal dynamics of these data. Such an approach could reveal risk factors that are not apparent in a conventional static analysis, thus providing a more comprehensive and integrated perspective of the factors influencing Ebola mortality during the follow-up period of participants in a clinical trial [9–11].The primary challenge addressed by our study lies in the limitations of current analytical approaches to comprehensively assess mortality risk factors in the context of longitudinal data that are often unbalanced, characterized by varying numbers of observations per patient due to differences in follow-up durations or incomplete data collection during Ebola clinical trials [12].

Many studies have employed traditional logistic regression models that are typically based on a single data collection point (often $Day_0$), overlooking

dynamic changes in patients' clinical conditions and variations in data panels over time. [13]. For instance, Loubet et al. (2016) developed a predictive model for EVD mortality using retrospective data from N'Zérékoré, Guinea, based on a single time-point analysis. Similarly, Levine et al. (2015) created a risk prediction score in Liberia using clinical and laboratory data at admission only. Yango et al. (2024) and Tshomba et al. (2022) also employed single-point predictive models in the DRC context, which constrained their ability to capture evolving risk patterns. While foundational, these models were unable to account for the temporal evolution of disease severity [8,14–17].

To address these gaps, our study employs a dynamic modeling framework that uses unbalanced panel data to evaluate changes in mortality predictors across multiple follow-up time points. This approach allows for a more detailed understanding of the disease trajectory and supports the implementation of timely and personalized interventions. It also accommodates the variability in follow-up observations common in clinical trials, where data are often incomplete or collected at irregular intervals [12,18].

The proposed statistical framework applies logistic regression iteratively at key follow-up visits, allowing us to identify mortality risk factors as they evolve over time. By integrating the temporal dimension, our approach maximizes the clinical insights gained from ongoing patient monitoring and offers a significant improvement over static model [18].

The primary objective of this study is to model mortality risk factors associated with Ebola Virus Disease (EVD) using binary logistic regression applied to unbalanced panel data from a phase II clinical trial. The specific aims are:

- To develop a statistical model tailored for the analysis of unbalanced panel data, accounting for variability across individuals, variables, and time in clinical trials.

- To apply this model to sequentially evaluate mortality risk factors in EVD patients enrolled in the PALM-RCT during the 10th epidemic in the Democratic Republic of Congo (DRC).

- To identify key mortality predictors that vary across follow-up visits, enabling the optimization of clinical care strategies and improving patient survival rates.

## Methods

### Ethics statement

The PALM-RCT clinical trial (ClinicalTrials.gov ID: NCT03719586) received ethical approval from the Ethics Committee of the Kinshasa School of Public Health (Approval No. ESP/CE/129/2018, issued November 20, 2018). Our secondary analysis of anonymized data from the PALM-RCT trial obtained ethical approval from the same committee (Approval No. ESP/CE/088/2023, issued January 11, 2023). An extension for data access was subsequently approved on January 15, 2024 (Approval No. ESP/CE/047/2024), allowing data use from January 15, 2024, to January 15, 2026.

The original data were collected during the 10th Ebola virus disease outbreak in the DRC (2018–2020) following ethical guidelines. Since the data were fully anonymized before analysis, the Ethics Committee waived the requirement for individual patient consent.

### Study design

This study is a retrospective secondary analysis of data from the PALM-RCT phase II clinical trial *(ClinicalTrials.gov ID: NCT03719586)* conducted during the 10th Ebola virus epidemic in the DRC, which evaluated the efficacy of several experimental treatments for the Ebola virus. We are using a retrospective cohort approach based on data collected during longitudinal visits of patients included in the PALM-RCT trial. These data include clinical, demographic, and biological variables collected at different times during the follow-up of Ebola-infected patients [2]

## Study population

The study population includes all patients enrolled in the PALM-RCT clinical trial, who were followed longitudinally during the epidemic. The study was carried out in three localities in the east of the Democratic Republic of Congo, in the provinces of Ituri and North Kivu, specifically Beni, Butembo Mangina and Katwa.

### Inclusion and exclusion criteria

- The primary inclusion criterion was a confirmed diagnosis of Ebola virus infection.

- Patients of any age, including pregnant women, with a positive RT-PCR result within 3 days prior to screening and a history of infection were eligible.

## Data collection

The data used in this analysis were obtained from the PALM-RCT clinical trial databases. These data include:

- Demographic information (age, gender, place of residence, contact details, Ebola vaccination status)

- Clinical data (symptoms, severity score, co-morbidities)

- Biological data (haematological and biochemical markers, viral load)

- Longitudinal follow-up data (clinical status at each follow-up visit).

Each visit date was treated as a panel, with the number of participants varying across panels, resulting in imbalanced data. Participants were followed up at multiple time points in accordance with the PALM-RCT trial protocol: at admission (Day 0), and subsequently on Days 7, 14, and 28, with an optional visit on Day 21. The primary endpoint for the study was mortality on Day 28.

Clinical, demographic, and biological data collected at these time points were used to analyze changes in mortality risk factors throughout the follow-up period. Data for this analysis were accessed for research purposes on ***04/02/2024.***

## Statistical analysis

### Data description.

- The data for this study were initially collected and organized in Microsoft Excel 2020 sheets. For statistical analysis, we used SPSS IBM Statistics version 27 software.

- The variable of interest was death (binary variable). Chi-square tests for categorical variables were used to test the association between each potential predictor and the dependent variable. For quantitative variables, a t-test (Student's t-test) was performed. The Kruskal-Wallis test was used to assess mortality in the different groups (panels), and the significance level used for all analyses was 5% ($p < 0.05$);

- The Area under curve (AUC) was calculated at each visit to assess the model's ability and the Hosmer-Lemeshow tests were used to calibrate the model and validate it internally.

- The study included 617 participants, spread over the different follow-up periods: $Day_0$ = 617 (inclusion), $Day_7$ = 389, $Day_{14}$ = 274, $Day_{21}$ = 81, and $Days_{28}$ = 142.

## Methods development

Logistic regression is a widely utilized statistical method in health research for identifying risk factors associated with specific outcomes and estimating the probability of an event based on independent variables. [13,19]. For this study, we employed a top-down elimination approach, specifically the step-by-step Wald method, to build the predictive model. This

iterative process selects the best-fit model by evaluating the statistical significance of each variable at each step, ensuring that only the most relevant predictors are retained [20]].

The top-down elimination process begins by including all available explanatory variables in the model. At each step, the Wald statistic is used to assess the significance of each variable, and the variable with the least statistical significance (p-value exceeding a predefined threshold, typically 0.05) is removed. This iterative evaluation and elimination continue until all remaining variables meet the significance criteria ($p < 0.05$).

The Wald statistic quantifies the relative importance of each variable, ensuring a rigorous selection process that results in a final model comprising only variables with substantial contributions to predicting the outcome. This method enhances model precision and interpretability by excluding non-influential variables and focusing on the strongest predictors.

The process stops when all the remaining variables in the model are significant, and no further variables can be added. [19,21].

In our case, we apply this method to panel data, where each patient visit constitutes a separate panel. We analyzed a total of five panels corresponding to the visits ($panel_1$ = $visit_0$ = $Day_0$ = day of inclusion, $panel_2$ = $visit_1$ = $Day_7$, $panel_3$ = $visit_2$ = $Day_{14}$, $panel_4$ = $visit_3$ = $Day_{21}$ et $panel_5$ = $visit_4$ = $Day_{28}$). $Y_t$ takes values in {0,1}, as a function of the explanatory variables $X_i$ at each time point t (visit time). The logistic model proposes to model the distribution of $Y_t/X_{i,t} = x_t$ by a Bernoulli distribution with parameter:

$$p_\beta(x_t) = P(Y_t = 1 | X_{i,t} = x_t) \tag{1.1}$$

Such as:

$$\log\left(\frac{p_{\beta(x_t)}}{1 - p_{\beta(x_t)}}\right) = \beta_0 + \beta_1 x_{1,0} + \ldots + \beta_i x_t = x'_t \beta$$

or

$$\text{logit } p_\beta(x_t) = x'_t \beta.$$

*logit* Denoting the bijective and differentiable function of $]0, 1[$ in $\mathbb{R} \longmapsto \log(p/(1-p))$.
The equality in (1.1) can be written as:

$$p_\beta(x_t) = P(Y_t = 1 | X_{i,t} = x_t) = \frac{\exp(x'_t \beta)}{1 + \exp(x'_t \beta)} \tag{1.2}$$

*Coefficients for estimating the odds ratio* $(\exp \beta)$.

In our study, the exogenous variables are not always binary and change over time. We obtain the odds ratio (OR) using the following formula:

$$RC = \frac{\frac{P(Y_t=1|X_{i,t}=1)}{1-P(Y_t=1|X_{i,t}=1)}}{\frac{P(Y_t=|X_{i,t}=0)}{1-P(Y_t=|X_{i,t}=0)}} \tag{1.3}$$

By the equality of (1.1) we have:

$$RC = \frac{\exp(\beta_0 + \beta_i)}{\exp(\beta_0)} = \exp \beta_i \tag{1.4}$$

The estimator $\beta_i$ gives the odds ratio when $X_{i,t}$ increases by one unit.

(a) Calibration and adjustment:

We use the Hosmer-Lemeshow test, a statistical test that has been used to assess the fit and calibration of the logistic regression model. It compares observed and predicted values to determine whether there are significant differences between them, which indicates goodness of fit and internal model validation Statistically the test consists of dividing the sample into groups (usually 10) according to estimated probabilities, then calculating a $\chi^2$ from the differences between observed and expected results in each group [19,20].

The $\chi^2$ is calculated by the following formula:

$$\chi^2 = \sum_{i=1}^{g} \frac{(O_i - E_i)^2}{E_i}$$

(1.5)

$O_i$ are the observed observations, $E_i$ are the expected observations, and $g$ is the number of groups. In our case, we will carry out the test at each visit time, and we will obtain:

$$\chi^2{}_t = \sum_{i=1}^{g} \frac{(O_{it} - E_{it})^2}{E_{it}}$$

(1.5)

Where $t = Day_0, Day_7, Day_{14}, Day_{21}, Day_{28}$

The p-value returned by the test statistic must be greater than 5% (significance level) to indicate acceptable agreement.

## Results

In our study, 617 participants were recruited, all of whom met the inclusion criteria, defined as a positive PCR test result for EVD. These participants were subsequently allocated to different visit dates for follow-up (Fig 1).

The demographic, biochemical, vital, and clinical characteristics of the participants, summarized in Table 1, indicate that the mean age of participants was 29 ± 18 years, with a consistent across the visits with a mean age around 29 years. The proportion of male participants was 55.58%, decreasing slightly from 55.58% on Day0 to 46.29% on Day21. Biochemically, viral load increased from 24.22 on Day0 to 37.61 on Day21, with a slight decrease to 34.54 on Day28. Creatinine levels fell from 2.64 mg/dL on Day0 to 0.62 mg/dL on Day14, reflecting improved renal function. Sodium levels remained stable at approximately 137.50 mEq/L, while liver enzymes aspartate aminotransferase (ASAT) and alanine aminotransferase (ALAT) showed significant declines, indicating enhanced liver function.

Vital signs revealed a gradual increase in systolic blood pressure (from 100.14 to 117.73 mmHg) and diastolic blood pressure (from 63.55 to 75.4 mmHg) between Day 0 and Day28. Heart rate stabilized at around 95 beats per minute, with a slight dip on Day 7. Body temperature and respiratory rate showed slight decreases over the course of the study. Clinically, symptoms such as fever (312 cases on Day 0–4 cases on Day 28), cough, headache, vomiting, diarrhea, abdominal pain, and conjunctival injection decreased significantly, demonstrating marked clinical improvement in participants by Day 28. Data on mortality according to patient follow-up, presented in Fig 2, show a decreasing trend over time. 265 deaths were recorded at the inclusion ($Day_0$), which represent 43% of all the deaths. At $Day_7$, mortality fell sharply to 27 deaths (7–4%), indicating a rapid stabilization of the survival status. Less than 1,82% of deaths occurred on and after $Day_{14}$, suggesting a consolidation of the positive clinical results. The Kruskal-Wallis test (p-value = 0.002) showed that the reduction in mortality in the different groups (panels) was significant at the 5% threshold.

We used data from inclusion (n = 617) to find relationships between different variables in the model. The table 2 shows that certain clinical and biochemical characteristics are strongly associated with an increased risk of death. Important features such as signs of haemorrhage, coma and laboratory test results are important indicators that require special attention in patient management (see Table 2 for detailed results)..

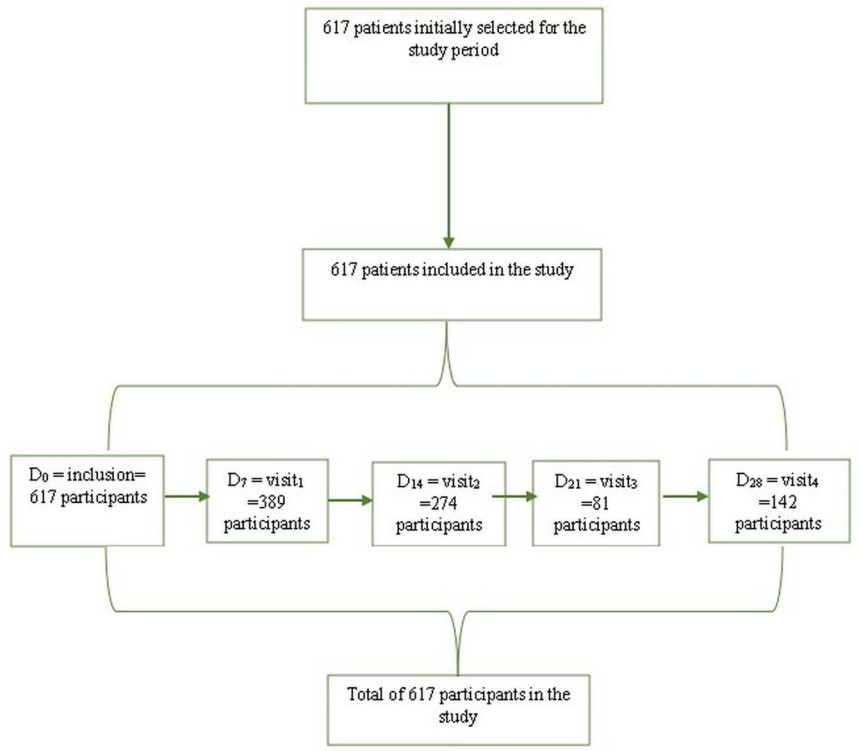

**Fig 1. The flow chart illustrates the process of selecting participants for a randomized controlled trial on Ebola Virus Disease (EVD).**

The risk factors associated with mortality at different follow-up periods, presented in Table 3, vary according to the time of assessment. At $Day_0$(Model$_1$), six factors were associated with death: Viral load 0.78 (0.72-0.85), Creatinine 1.18 (1.08-1.29), ALAT 1.00 (1.01-1.03), Hemorrhage 3.81 (1.97-7.35), Shortness of breath 2.99 (1.21-7.39) and Conjunctivitis 1.95 (1.09-3.50). By Day7 (Model 2), five predictors were identified: sodium 1.09 (1.03–1.17), ASAT 1.00 (1.00–1.01), coma 8.40 (2.22–31.77), abdominal pain 4.30 (1.33–13.90), and shortness of breath 4.88 (1.55–15.38). Notably, shortness of breath persisted as a significant factor from Day0, while the other predictors either newly emerged or became significant at this later stage. At Day14 (Model 3), two predictors remained associated with mortality: ASAT 1.03 (1.01–1.05) and coma 57.05 (1.25–2612.16). These findings highlight the dynamic evolution of risk factors over time, with some predictors persisting while others vary across different stages of disease progression.

The performance of the models, illustrated in Fig 3, varies across the follow-up periods. Model 1 demonstrates good performance with an area under the curve (AUC) of 0.88, indicating a risk factor-related probability of death of 88%. Similarly, Model 2 shows strong performance with an AUC of 0.85, reflecting an 85% probability of death among individuals with associated risk factors. Model 3 performs exceptionally well, achieving an AUC of 0.97, suggesting that individuals with the risk factors identified in this model have a greater than 97% likelihood of mortality. In contrast, Models 4 and 5, with AUCs of 0.50 each, exhibit poor predictive ability and fail to effectively distinguish between individuals at risk of death and those not at risk.

The evaluation of models at different time points, as shown in Fig 4, highlights the evolution of predictors influencing mortality. Model 1, based on data at inclusion, identified six significant predictors: elevated viremia (nucleoprotein Ct value), renal dysfunction (elevated creatinine), liver dysfunction (elevated alanine aminotransferase (ALAT)), hemorrhage, shortness of breath, and conjunctival infection. At Day7, Model 2 identified five predictors of mortality: electrolyte

**Table 1. Demographic, biochemical and clinical characteristics.**

| Characteristic | All, Day$_0$ (n=617) | Day$_7$ (n=389) | Day$_{14}$ (n=274) | Day$_{21}$ (n=82) | Day$_{28}$ (n=142) |
|---|---|---|---|---|---|
| Age-yr (Mean ±standard-deviation) | 29±18 | 28±18 | 29±17 | 22±16 | 29±20 |
| **Sex (n (%))** | | | | | |
| Male | 343(55.58) | 210(53.96) | 150(54.68) | 38(46.29) | 71(50.39) |
| Female | 274(44.42) | 179(46.40) | 124(44.32) | 44(53.71) | 70(49.59) |
| **Biochemical parameters (Mean ±standard-deviation)** | | | | | |
| Nucleoprotein Ct value | 24.22±4.21 | 31.83±10.29 | 33.62±9.21 | 37.61±6.87 | 34.54±5.29 |
| Creatine-mg/dl | 2.64±2.80 | 1.01±1.31 | 0.62±0.33 | 0.62±0.32 | 0.63±0.21 |
| Potassium-mmol/liter | 4.33±0.91 | 4.33±1.14 | 4.72±0.92 | 4.71±1.03 | 4.34±0.71 |
| Sodium_cu | 132.21±5.74 | 138.23±5.14 | 137.52±4.91 | 137.05±5.31 | 137.50±3.71 |
| ASAT-U/liter | 639.32±532.91 | 129.72±228.93 | 57.40±33.91 | 88.40±21.34 | 44.41±46.73 |
| ALAT-U/liter | 383.53±435.82 | 86.25±70.03 | 50.22±36.84 | 54.20±84.23 | 33.50±25.71 |
| **Vital signs (Mean ±standard-deviation)** | | | | | |
| Blood pressure-mm Hg | | | | | |
| Systolic | 100.14±32.40 | 114.33±17.03 | 113.11±13.81 | 112.60±14.03 | 117.73±15.72 |
| Diastolic | 63.55±21.90 | 72±17.03 | 73.74±12.11 | 71.60±12.91 | 75.44±12.73 |
| Pulse-beats/min | 97.20±22.90 | 92.30±12.50 | 93.4±16.9 | 99.50±23.30 | 95.82±16.60 |
| Body temperature -C | 37.39±1.79 | 36.69±0.92 | 36.53±0.71 | 36.51±0.74 | 36.32±0.52 |
| Respiratory rate-breaths/min | 26.52±8.70 | 23.83±7.04 | 22.12±5.12 | 220±3.92 | 21.31±3.13 |
| Oxygen saturation | 94.92±7.91 | 97.32±2.81 | 97.63±2.42 | 98.42±1.50 | 97.66±1.83 |
| **Clinical signs (n, %)** | | | | | |
| Fever | 312(50.56) | 62(15.93) | 17(6.20) | 5(6.09) | 4(2.81) |
| Cough | 61(9.88) | 20(5.14) | 8(2.91) | 1(1.22) | 3(2.11) |
| Headache | 275(44.57) | 23(5.91) | 9(3.28) | 2(2.44) | 16(11.27) |
| Vomiting | 237(38.41) | 23(5.91) | 2(0.72) | 1(1.22) | 0(0.00) |
| Diarrhea | 324(52.51) | 73(18.76) | 14(5.11) | 1(1.22) | 1(0.70) |
| Haemorrhage | 92(14.91) | 14(3.59) | 0(0.00) | 0(0.00) | 1(0.70) |
| Convulsions | 10(1.62) | 4(1.03) | 0(0.00) | 0(0.00) | 2(1.41) |
| Coma | 30(4.86) | 15(3.85) | 4(1.46) | 0(0.00) | 2(1.41) |
| Abdominal | 283(38.57) | 31(7.96) | 2(0.72) | 2(2.44) | 6(4.23) |
| Shortness of breath, | 44(7.13) | 26(6.63) | 4(1.46) | 0(0.00) | 0(0.00) |
| Hiccups | 19(3.08) | 4(1.03) | 0(0.00) | 0(0.00) | 0(0.00) |
| Rash | 11(1.78) | 5(1.28) | 2(0.72) | 0(0.00) | 1(0.70) |
| Conjunctival injection | 110(17.82) | 19(4.88) | 5(1.82) | 0(0.00) | 1(0.70) |

*ASAT*: Aspartate aminotransferase.

*ALAT*: Alanine aminotransferase.

imbalances (sodium), liver dysfunction (elevated aspartate aminotransferase (ASAT)), central nervous system dysfunction (Coma), abdominal pain, and shortness of breath. By Day14, Model 3 retained two key predictors: persistent liver dysfunction (elevated ASAT) and central nervous system dysfunction (Coma).

The calibration and fit of our logistic regression models were assessed using the Hosmer-Lemeshow test (see Table 4 for detailed results). The results indicate that for Day0, Day7, and Day14, the models demonstrated a good fit to the data, as evidenced by high p-values exceeding the threshold of 0.05. These findings suggest that the conclusions drawn from

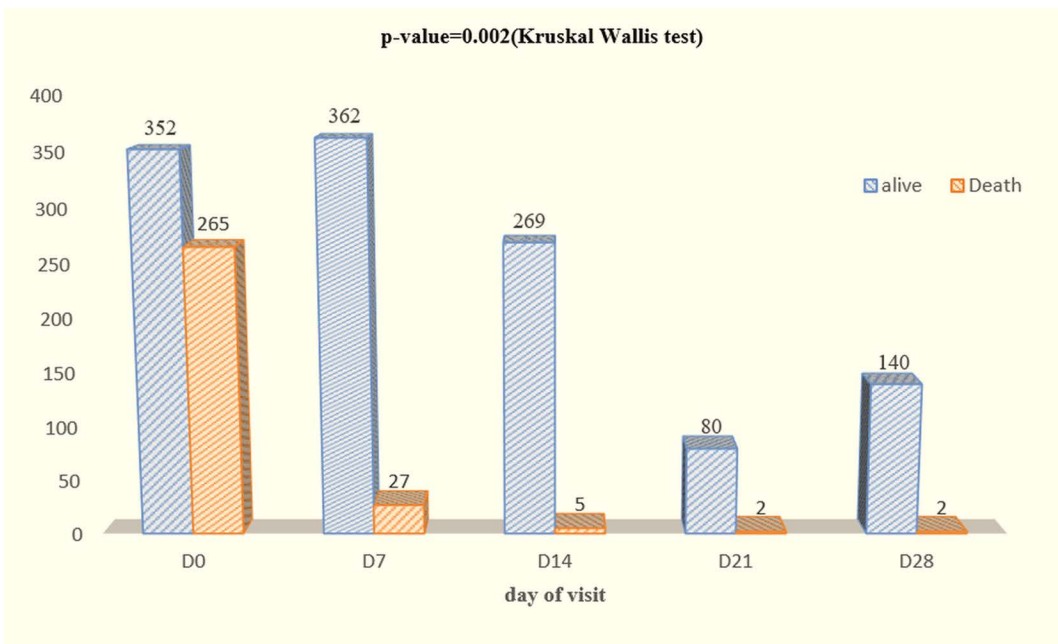

**Fig 2. *Mortality by group (follow-up time).***

these three models are robust and reliable. Conversely, the models for Day21 and Day28 exhibited poor fit, with p-values below the acceptable threshold, rendering them invalid for drawing reliable conclusions.

## Discussion

The aim of our study was to model the risk factors associated with mortality in patients with Ebola virus disease, using logistic regression applied to unbalanced longitudinal data. In the epidemiological context of Ebola virus infection, this approach captures variations in predictors of mortality over time, reinforcing the importance of dynamic risk assessment to improve care. We identified key predictors at different time points during follow-up (Day$_0$, Day$_7$, and Day$_{14}$), highlighting the need for an evolutionary approach to assessing risk and adapting care optimally.

On Day$_0$, six factors proved significant: a high level of viremia (nucleoprotein Ct value), deterioration in renal function (high creatinine level), deterioration in liver function (high ALAT level), hemorrhage, shortness of breath and conjunctival injection. These results point to critical markers on admission that could justify aggressive management from the time of diagnosis. At Day$_7$, five other predictive factors emerged: electrolyte imbalances (sodium), deterioration in liver function (elevated ASAT), alterations in the central nervous system (coma), abdominal pain and shortness of breath. Finally, at Day$_{14}$, two predictive factors were identified: deterioration in liver function (elevated ASAT) and central nervous system alterations (coma).

The results of our study compare favorably with those of previous research, notably those of [2,7,8,16,21,22] had also observed the importance of viral load and liver function markers at the start of the disease, but their study did not take account of changes in risk factors over time.

Our study, on the other hand, has shown that the initial predictors at Day$_0$ evolve, and that parameters such as electrolyte imbalances (sodium) and neurological symptoms (coma) become more relevant after the first week of follow-up.

In contrast to static analyses, such as the studies by Loubet et al. (2016), Levine et al. (2015), Yango J et al. (2024) or Tshomba AO and al. (2022) and also Leader L et al.(2023), which focus solely on data at inclusion (Day$_0$), our dynamic

**Table 2. Relationship between sociodemographic, clinical and biochemical factors and participant outcome.**

| Characteristic | Deaths | | p-value |
| --- | --- | --- | --- |
| | No (n = 352) | Yes (n = 265) | |
| **Socio-demographic characteristics** | | | |
| dm_sex (n (%)) | | | 0.78 * |
| Male | 194(56.59) | 149(43.41) | |
| Female | 158(57.680) | 116(42.32) | |
| Age (Mean ±standard-deviation) | 29.03 ± 18.01 | 30.04 ± 18.03 | 0.41** |
| **Clinical sign present (n (%))** | | | |
| Fever | 178(57.12) | 134(42.88) | 0.99* |
| Cough | 30(49.19) | 31(50.81) | 0.19* |
| Headache | 167(60.64) | 108(39.26) | 0.98* |
| Vomiting | 118(49.56) | 119(50.44) | 0.004* |
| Diarrhea | 167(51.56) | 157(48.44) | 0.004* |
| Haemorrhage | 17(18.77) | 75(81.23) | 0.00* |
| Convulsions | 3(30.05) | 7(69. 95.00) | 0.81* |
| Coma | 7(23.31) | 23(76.09) | 0.00* |
| Abdominal pain | 156(55.15) | 127(44.95) | 0.37* |
| Shortness of breath, | 11(25.04) | 33(74.96) | 0.00* |
| Hiccups | 3(15.65) | 16(84.35) | 0.00* |
| Rash | 5(45.57) | 6(54.43) | 0.43* |
| Conjunctival injection | 38(34.43) | 72(65.57) | 0.00* |
| **Biochemical parameters (Mean ±standard-deviation)** | | | |
| Nucleoprotein Ct value | 25.76 ± 4.02 | 22.10 ± 3.42 | 0.00** |
| Creatinine | 1.74 ± 2.14 | 3.67 ± 3.10 | 0.00** |
| Potassium | 4.18 ± 0.88 | 4.53 ± 0.96 | 0.00** |
| Sodium | 132.83 ± 536.12 | 131.48 ± 5.88 | 0.004** |
| ASAT | 510.64 ± 33.91 | 810.07 ± 478.89 | 0.00** |
| ALAT | 197.07 ± 226.14 | 632.12 ± 517.17 | 0.00** |

** Student's T test

* Chi-square test

analysis shows that risk factors evolve over time and that recognizing this evolution is critical for effective management. [23] studies have emphasized the importance of patient-centered approaches, and our results reinforce this idea by demonstrating that personalizing care according to patients' clinical evolution is crucial for improving outcomes [2,7,8,22]. The results of this study have important implications for the clinical management of patients with Ebola virus disease.

The identification of different predictors at each stage of follow-up means that therapeutic interventions can be better targeted. For example, viral load and markers of renal function (creatinine) are critical indicators on admission, justifying aggressive treatment from the moment of diagnosis. On Day$_7$, the appearance of factors such as changes in mental status and sodium underlines the importance of neurological and biochemical monitoring to anticipate potential deterioration. Finally, on Day$_{14}$, the persistence of elevated liver enzymes (ASAT) and neurological symptoms may require more intensive management of vital functions.

These results suggest that dynamic assessment of Ebola patients could lead to more personalized management, where treatments are adjusted according to changes in risk over time, and not just based on initial patient characteristics. It could also help optimize the use of medical resources, particularly in resource-poor settings, by targeting higher-risk

PLOS Global Public Health

**Table 3. Assessment of risk factors using logistic regression.**

| Variables in the equation | B | E. S | Wald | ddl | p-value | Odd-Ratio/ (Exp(B)) | 95% confidence interval for EXP(B) | |
|---|---|---|---|---|---|---|---|---|
| Model $e_1$ = Day$_0$ = Inclusion | | | | | | | Lower | Superior |
| Nucleoprotein Ct value | -0.25 | 0.04 | 38.13 | 1 | 0.00* | 0.78 | 0.72 | 0.85 |
| Creatine | 0.17 | 0.05 | 13.25 | 1 | 0.00* | 1.18 | 1.08 | 1.29 |
| ALAT | 0.00 | 0.00 | 31.3 | 1 | 0.00* | 1.00 | 1.01 | 1.03 |
| Headache | -0.39 | 0.22 | 3.11 | 1 | 0.08 | 0.68 | 0.44 | 1.04 |
| Haemorrhage | 1.34 | 0.34 | 15.84 | 1 | 0.00* | 3.81 | 1.97 | 7.35 |
| SOB breathing | 1.09 | 0.46 | 5.61 | 1 | 0.02* | 2.99 | 1.21 | 7.39 |
| Hiccups | 1.34 | 0.71 | 3.62 | 1 | 0.057 | 3.83 | 0.96 | 15.31 |
| Conjunctival injection | 0.67 | 0.29 | 4.99 | 1 | 0.03* | 1.95 | 1.09 | 3.50 |
| Constant | 4.15 | 0.98 | 17.876 | 1 | 0.00 | 63.16 | | |
| Model$_2$ = Day$_7$ | | | | | | | | |
| Sodium Cu | 0.09 | 0.03 | 7.36 | 1 | 0.01* | 1.09 | 1.03 | 1.17 |
| ASAT | 0.004 | 0.00 | 14.58 | 1 | 0.00* | 1.00 | 1.00 | 1.01 |
| ALAT | -0.01 | 0.01 | 3.53 | 1 | 0.06 | 0.99 | 0.98 | 1 |
| Coma | 2.13 | 0.67 | 9.84 | 1 | 0.00* | 8.40 | 2.22 | 31.77 |
| Abdominal pain | 1.46 | 0.598 | 5.95 | 1 | 0.02* | 4.30 | 1.33 | 13.90 |
| Shortness of breath | 1.59 | 0.586 | 7.32 | 1 | 0.01* | 4.88 | 1.55 | 15.38 |
| Constant | -15.63 | 4.665 | 11.22 | 1 | 0.001 | 0.00 | | |
| Model$_3$ = Day$_{14}$ | | | | | | | | |
| ASAT | 0.03 | 0.01 | 10.43 | 1 | 0.00* | 1.03 | 1.01 | 1.05 |
| Fever | 2.40 | 1.35 | 3.14 | 1 | 0.08 | 11.02 | 0.78 | 156.72 |
| Coma | 4.04 | 1.95 | 4.29 | 1 | 0.04* | 57.05 | 1.25 | 2612.16 |
| Constant | -7.69 | 1.56 | 24.29 | 1 | 0.00 | 0.00 | | |

*B: Regression coefficient.*

*E. S: (Standard error): This is a measure of the precision of the estimate of the B coefficient.*

*Wald: The Wald test is used to determine whether the B coefficient is statistically significant.*

*ddl (degrees of freedom): This refers to the number of free values in a calculation.*

*95% confidence interval for EXP(B): This interval indicates the range within which we can be 95% sure that the true value of Exp(B) lies.*

*\*: Significative P-value.*

patients at different stages of the disease. A major strength of our study is the use of unbalanced longitudinal data, which has allowed us to capture the dynamics of risk factors throughout the course of the disease. This approach offers a better understanding of the clinical changes that occur over time in Ebola-infected patients, and we believe that this reinforces the clinical value of our results.

## Conclusion

Our study successfully modeled mortality risk factors in patients with EVD using logistic regression on unbalanced longitudinal data. The results identified significant predictors at different stages of clinical follow-up (Day0, Day7, and Day14), underscoring the critical importance of monitoring changes in risk factors over time to adapt therapeutic strategies. For instance, viral load and renal and hepatic function emerged as key predictors at admission, whereas changes in mental status and electrolyte imbalances became more prominent at later stages of the disease. These findings highlight the potential of personalized care tailored to patients' evolving clinical profiles to significantly improve therapeutic outcomes

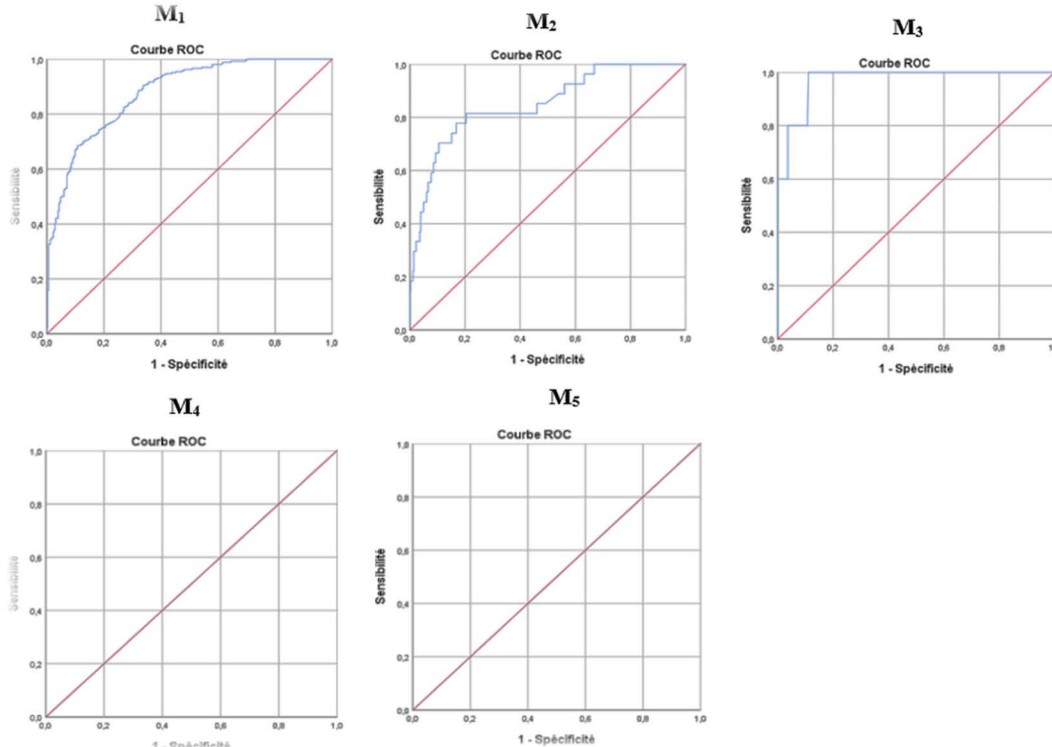

**Fig 3.** *Model performance represented by receiver operating characteristic (ROC) curves across follow-up periods.*

## KEY RISK FACTORS ASSOCIATED WITH MORTALLITY ON MONITORING TIME

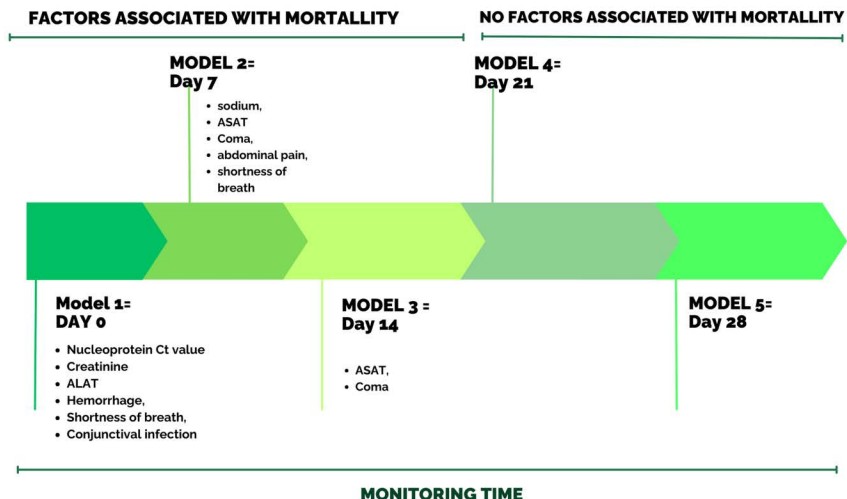

**Fig 4.** *Key risk factors associated with mortality.*

**Table 4. Hosmer-Lemeshow Test Results for Model Calibration Across Time Points.**

| Model | Chi-square | ddl | p-value |
|---|---|---|---|
| $Day_0$ | 18.44 | 8 | 0.82 |
| $Day_7$ | 2.60 | 8 | 0.96 |
| $Day_{14}$ | 2.68 | 8 | 0.95 |
| $Day_{21}$ | – | – | – |
| $Day_{28}$ | – | – | – |

***ddl (degrees of freedom):*** *This refers to the number of free values in a calculation.*

and reduce mortality. Continuous monitoring and real-time adjustments in clinical management could help optimize care and mitigate severe complications effectively.

Finally, prospective multicenter studies are essential to validate these findings and support the integration of this dynamic risk assessment model into routine clinical practice, further enhancing patient outcomes in future EVD outbreaks.

## Limitation

This study has several limitations. First, as a retrospective secondary analysis, the findings may not be directly generalizable to future outbreaks without further validation. Additionally, the dynamic nature of EVD and variability across clinical settings require cautious interpretation and underscore the importance of external prospective validation in diverse geographic and epidemiologic contexts. Finally, the unbalanced nature of the dataset due to differences in follow-up durations and patient outcomes may affect statistical power at later time points, potentially limiting the generalizability of visit-specific results despite the use of appropriate analytical methods.

## Future studies should consider

• Conducting prospective, multicenter cohort studies to externally validate these findings.

• Integrating dynamic risk prediction models into routine clinical workflows for real-time decision-making.

• Investigating the longitudinal evolution of clinical and biochemical markers to refine disease staging and therapeutic strategies.

## Supporting information

**S1 Data. PALM Dataset.**
(XLSX)

## Acknowledgments

We extend our heartfelt gratitude to the researchers from the Department of Epidemiology and Global Health and the Department of Immunology at the National Institute for Biomedical Research for their invaluable contributions to the discussions during the scientific days. We also sincerely thank the PALM-RCT consortium authorities, particularly Dr. Jean-Luc Biampata, for generously providing the data that made this article possible.

## Author contributions

**Conceptualization:** Leader Lawanga Ontshick, Jepsy Yango, Rosine Ali.

**Data curation:** Leader Lawanga Ontshick, Jepsy Yango, Ange Mubiala Yaya, Patrick Mutombo Lupola.

**Formal analysis:** Leader Lawanga Ontshick, Jepsy Yango, Ange Mubiala Yaya, Rostin Mabela Makengo Matendo.

**Funding acquisition:** Leader Lawanga Ontshick, Jepsy Yango.

**Investigation:** Leader Lawanga Ontshick, Jepsy Yango.

**Methodology:** Leader Lawanga Ontshick, Jepsy Yango, Ange Mubiala Yaya, Patrick Mutombo Lupola, Sifa Marie-joelle Muchanga.

**Project administration:** Leader Lawanga Ontshick, Jepsy Yango.

**Resources:** Leader Lawanga Ontshick, Jepsy Yango.

**Software:** Leader Lawanga Ontshick, Jepsy Yango, Ange Mubiala Yaya, Patrick Mutombo Lupola.

**Supervision:** Leader Lawanga Ontshick, Jepsy Yango, Joseph-Désiré Bukweli, Placide Mbala Kiangebeni, Sabue Mulangu, Rostin Mabela Makengo Matendo.

**Validation:** Leader Lawanga Ontshick, Jepsy Yango, Jean-Michel Nsengi Ntamabyaliro, Gaston Tona Lutete, Sabue Mulangu, Rostin Mabela Makengo Matendo.

**Visualization:** Leader Lawanga Ontshick, Jepsy Yango, Sabue Mulangu, Rostin Mabela Makengo Matendo.

**Writing – original draft:** Leader Lawanga Ontshick, Jepsy Yango, Jean-Michel Nsengi Ntamabyaliro, Rosine Ali, Sifa Marie-joelle Muchanga, Gaston Tona Lutete, Placide Mbala Kiangebeni, Sabue Mulangu, Rostin Mabela Makengo Matendo.

**Writing – review & editing:** Leader Lawanga Ontshick, Jepsy Yango, Olivier Tshiani Mbaya, Joule Madinga Twan, Jean-Michel Nsengi Ntamabyaliro, Rosine Ali, Joseph-Désiré Bukweli, Sifa Marie-joelle Muchanga, Gaston Tona Lutete, Placide Mbala Kiangebeni, Sabue Mulangu, Rostin Mabela Makengo Matendo.

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
