## [Decision Letter · Decision Letter 0]

PGPH-D-24-02895

Dynamic Modeling of Mortality Risk Factors in Ebola Virus Disease Using Logistic Regression on Unbalanced Panel Data from a Randomized Controlled Trial in the Democratic Republic of Congo

Dear Mr. Leader Lawanga Ontshick,

Thank you for submitting your manuscript to PLOS Global Public Health. After careful consideration, we feel that it has merit but does not fully meet PLOS Global Public Health’s publication criteria as it currently stands. Therefore, we invite you to submit a revised version of the manuscript that addresses the points raised during the review process.

We look forward to receiving your revised manuscript.

Kind regards,

Dr Vishal Goyal

Academic Editor

Journal Requirements:

1. We have noticed that you have uploaded Supporting Information files, but you have not included a list of legends. Please add a full list of legends for your Supporting Information files after the references list.

**Comments to the Author**

Reviewer #1: 

Reviewer’s Comments of PGPH-D-242895

The authors studied “Dynamic Modeling of Mortality Risk Factors in Ebola Virus Disease Using Logistic Regression on Unbalanced Panel Data from a Randomized Controlled Trial in the

Democratic Republic of Congo." The findings highlight the potential of personalized care tailored to patients' evolving clinical profiles to significantly improve therapeutic outcomes and reduce mortality. Continuous monitoring and real-time adjustments in clinical management could help optimize care and mitigate severe complications effectively. The work seems to be good.

However, in my view, the authors need to consider the following suggestions (comments) to improve this article.

1. There are some grammatical errors in the manuscript. I think it needs to be polished further, and some typos need to be revised. Further, punctuation marks should be checked throughout the paper, especially after the equations. Authors need to correct them throughout the manuscript. For instance,

i. In line 232, there should be a full stop behind the validation. And all other typos should be corrected

2. All equations must be well punctuated, centered and typed well. For instance, equations in lines 214, 216, 218 should be well. The fraction in line 222 must be typed in the form A=D/C.

3. Authors should clearly state their motivation for this research in the introduction section.

4. According to the authors at lines 118 and 119, “Many studies have employed traditional logistic regression models that are typically based on a single data collection point…” Yet, they cited only two papers at the review or related work section. Authors should expand the review of literature, compare their current research to the existing literature, and clearly state the innovation in this research in the introduction section.

5. Authors should indicate the source of the parameter values used for their simulations. If taken from literature, then they should cite the papers in the current study.

6. Authors should indicate limitations of their research

7. Authors should indicate the future direction of their research.

8. Authors should state the implication of their findings/results

9. Titles/ Labels of all figures(graphs) should be given.

Reviewer #2: At D0, six significant predictors of mortality were determined, although seven are listed.

At Day7, reference is made to five new predictors. Not all of these are entirely new, as two were previously identified at D0. Would suggest rewording the use of “new predictors”

Line 155: Not sure the significance of it. It is confusing

Line 206: Shouldn’t read “further variables can be removed” because of the top-down elimination process?

Linke 245: Update PALM to “PALM-RCT”

Line 265: Consider using “age was consistent across the visits…” vs “consistent distribution between 22 and 29 years of age”

Ensure consistent use of AST vs ASAT, ALT vs ALAT, Haemorrhage (check consistent and spelling throughout the document), Difficulty breathing vs SoB, …

Line 268: Creatine levels remained at 0.6 at D21 and D28.

Lines 283 – 285: Re-calculate the percentages. For example, 265 deaths of 617 = 43%?

Table 2: Please show the number of participants associated with the Biochemical parameters.

Was Mental state captured at D0? Would be good to know what its inclusion means.

Was Generalized Linear Mixed Model considered for the modelling?

I think we could describe the statistical methods clearly without the formulas.

---

## [Decision Letter · Decision Letter 1]

PGPH-D-24-02895R1

Dynamic Modeling of Mortality Risk Factors in Ebola Virus Disease Using Logistic Regression on Unbalanced Panel Data from a Randomized Controlled Trial in the Democratic Republic of Congo

Dear Mr Leader Lawanga Ontshick,

Thank you for submitting your manuscript to PLOS Global Public Health. After careful consideration, we feel that it has merit but does not fully meet PLOS Global Public Health’s publication criteria as it currently stands. Therefore, we invite you to submit a revised version of the manuscript that addresses the points raised during the review process.

We look forward to receiving your revised manuscript.

Kind regards,

Dr Vishal Goyal

Academic Editor

Journal Requirements:

**Reviewers Comments to the Author**

Reviewer #1: Reviewer’s Comments (PGPH-D-24-02895R1)

The authors studied the Dynamic Modeling of Mortality Risk Factors in Ebola Virus Disease Using Logistic Regression on Unbalanced Panel Data from a Randomized Controlled Trial in the

Democratic Republic of Congo.

I have checked the revised version of the manuscript according to the comments given, but I still have a few suggestions to be addressed by the authors. These are:

Comment 1: Keywords

Authors should either use ‘’EVD’’ or Ebola Virus Disease but not Ebola Virus Disease(EVD) within the Keywords.

Comment 2: Equation formatting

All equations involving fractions in the manuscript should be written in the form A= B/C (B/C should be vertical), but not (A=B)/C. Authors should please address that.

Comment 2: Limitations and Future direction

a. Limitations and Future directions should be part of the conclusion. Authors should summarize them and add them to the ending section of the conclusion, just before the references.

b. Limitations should be stated first before suggestions for future considerations.

c. Suggestions for future considerations should be stated in points just before the references.

Conclusion: The manuscript can be accepted after these minor revisions have been made to be published in PLOS Global Public Health.

Reviewer #2: There are some occurrences of both PALM001-RCT and PALM-RCT. Please review this for consistency.

---

## [Editor Report · Decision Letter 2]

Dynamic Modeling of Mortality Risk Factors in Ebola Virus Disease Using Logistic Regression on Unbalanced Panel Data from a Randomized Controlled Trial in the Democratic Republic of Congo

PGPH-D-24-02895R2

Dear Mr Leader Lawanga Ontshick, 

We are pleased to inform you that your manuscript 'Dynamic Modeling of Mortality Risk Factors in Ebola Virus Disease Using Logistic Regression on Unbalanced Panel Data from a Randomized Controlled Trial in the Democratic Republic of Congo' has been provisionally accepted for publication in PLOS Global Public Health.

Best regards,

Vishal Goyal

Academic Editor
